# Modification of Iron-Rich Phase in Al-7Si-3Fe Alloy by Mechanical Vibration during Solidification

**DOI:** 10.3390/ma16051963

**Published:** 2023-02-27

**Authors:** Cuicui Sun, Suqing Zhang, Jixue Zhou, Jianhua Wu, Xinfang Zhang, Xitao Wang

**Affiliations:** 1Shandong Provincial Key Laboratory of High Strength Lightweight Metallic Materials, Advanced Materials Institute, Qilu University of Technology (Shandong Academy of Sciences), Jinan 250014, China; 2School of Metallurgical and Ecological Engineering, University of Science and Technology Beijing, Beijing 100083, China

**Keywords:** modification mechanism, iron-rich phases, mechanical vibration, Al-7Si-3Fe alloy

## Abstract

The plate-like iron-rich intermetallic phases in recycled aluminum alloys significantly deteriorate the mechanical properties. In this paper, the effects of mechanical vibration on the microstructure and properties of the Al-7Si-3Fe alloy were systematically investigated. Simultaneously, the modification mechanism of the iron-rich phase was also discussed. The results indicated that the mechanical vibration was effective in refining the α-Al phase and modifying the iron-rich phase during solidification. The forcing convection and a high heat transfer inside the melt to the mold interface caused by mechanical vibration inhibited the quasi-peritectic reaction: L + α-Al_8_Fe_2_Si → (Al) + β-Al_5_FeSi and eutectic reaction: L → (Al) + β-Al_5_FeSi + Si. Thus, the plate-like β-Al_5_FeSi phases in traditional gravity-casting were replaced by the polygonal bulk-like α-Al_8_Fe_2_Si. As a result, the ultimate tensile strength and elongation were increased to 220 MPa and 2.6%, respectively.

## 1. Introduction

Aluminum alloys are widely used in the transportation, aerospace, building, and packaging industries due to their lightweight properties, high specific strength, high corrosion resistance, and excellent recyclability [1,2,3]. In the aluminum industry, applying recycled raw materials reduces energy consumption by more than 95%. Recently, recycled aluminum alloys have been regarded as an alternative to crude aluminum under the background of carbon neutrality. Unfortunately, the Fe element is an inevitable and harmful impurity present in recycled aluminum alloys, and it is easily enriched during aluminum alloy recycling [4,5,6]. Various brittle iron-rich intermetallic phases such as Al_3_Fe (θ), Al_8_Fe_2_Si (α), and Al_5_FeSi (β) formed in the recycling process; among the plate-like or needle-shaped β-Al_5_FeSi phase (β phase) not only split matrix but also promote the formation of casting defects, such as pore and shrinkage porosity. These can act as locations of high-stress concentrations and severely deteriorate the performance of the recycled aluminum alloys [7,8,9]. However, the iron-rich phase is also an ideal strengthening phase for recycled aluminum, which results from its high hardness, good wettability with the matrix, and thermal stability [10]. Therefore, modifying the iron-rich phases will extend the application of recycled aluminum alloys.

With regard to reducing the disadvantages of plate-like iron-rich phases, many researchers have been interested in modifications. It has been proved that the morphology of the iron-rich phase can be modified by melt treatment [5,11], neutralizing elements addition (such as Mn, La) [12,13], rapid solidification process [14,15], and melt superheating [16,17]. Thus, the plate-like iron-rich phases are replaced by Chinese-script, star-like, or polygonal bulk-like phases, which lead to an increase in UTS and elongation of recycled aluminum. However, the addition of neutralizing elements may form coarse phases, which could reduce the properties of the alloy. For example, Song et al. [12] point out that when the Mn content was higher than 0.93% in the Al-7Si-1.2Fe alloy, the plate-like Fe-rich phase was eliminated, but the size of the star-like Fe-rich phase increased, which reduced the ductility. The research [13] shows La additions can refine effectively the β-Fe phase in Al-7Si-4Cu-0.35Mg-0.2Fe, while exceeds 0.15 wt.%, the β-Fe and La-rich particles are crack initiation sources and mainly affect the final failure. In addition, neutralizing elements addition can also affect the recycling of aluminum alloys. Meanwhile, the applications of rapid solidification and melt superheating are also limited because of their higher costs and complicated processes.

It is well known that melt treatment is a relatively simple method in comparison with other processing and can effectively improve the properties of alloy [5,11,18]. During solidification, the melt is usually treated by mechanical vibration, ultrasonic waves, electromagnetic fields, and so on. Among them, mechanical vibration treatment has attracted more and more attention, thanks to its simple and low cost without introducing a new element. It has been proven that [19,20,21,22,23] mechanical vibration can refine grains due to the promotion of nucleation, reduce shrinkage porosities due to improving metal feeding, modify the second phase, and produce a more homogenous metal structure. These improved features can effectively enhance mechanical properties and lower susceptibility cracks. For example, after applying mechanical vibration to A356 alloy, the coarser dendrites transformed into fine and uniform equiaxed grains, and eutectic Si particles were refined, which increased the tensile strength, yield strength, elongation, and hardness by 35%, 42%, 63%, and 29%, respectively [20]. When the mechanical vibration range of 30~50 Hz is applied to LM25 aluminum alloy, the inner casting defects reduce and become smaller, and the scattering of mechanical properties is enhanced [21]. The flaky structure of aluminum transforms to fibrous by applying mechanical vibration of 100 HZ and 149 μm to Al-12Si alloy [22]. However, it is not clear whether mechanical vibration during solidification will modify the iron-rich phases in the Al-Si-Fe alloy.

The aim of the present work was to investigate the effect of mechanical vibration on the morphology and size of α-Al and the iron-rich phases of Al-7Si-3Fe. In addition, the mechanism of mechanical vibration on the morphology of the iron-rich phase was discussed by OM, SEM, XRD, the Thermo-Calc software, and so on.

## 2. Experimental

The raw materials were pure Al, Al-20 wt.%Si, and Al-20 wt.%Fe master alloy. The amount of pure Al, Al-20 wt.%Si, and Al-20 wt.%Fe master alloy were melted in a graphite crucible in an electric resistance furnace under 750 °C, and then the melt was degassed for 10 min with argon gas at 720 °C. After holding for 10 min at a temperature of 680 °C, the melt was poured into a stainless-steel mould, which was installed on a mechanical vibration device. The schematic diagram of the mechanical vibration device is shown in Figure 1. Subsequently, the mechanical vibration device was launched until the melt was totally solidified. The vibration conditions were frequency (30 Hz), amplitude (0.6 mm), and direction (vertical). The as-vibration cast Al-7Si-3Fe alloy was obtained (marked as as-vibration). For comparison, some of the melt was solidified under the same conditions without mechanical vibration, and the traditional cast Al-7Si-3Fe alloy was obtained (marked as as-cast). The chemical compositions of as-vibration and as-cast alloys were analyzed by an X-ray fluorescence (XRF) analyzer, as shown in Table 1.

The solidification sequences of the Al-Si-Fe ternary system alloy were calculated using the TCAL4 database within the Thermo-Calc software with the assumption of equilibrium solidification under the condition of complete diffusion and redistribution of all solute atoms. The cooling curves of the alloy were measured by a thermocouple and datalogger. The thermocouple fixed by the holder was inserted into the melt during solidification and connected to the datalogger, which was used to record the data. Especially, the thermocouples were calibrated against the melting of pure aluminum, and the datalogger included a temperature measurement module and data recording software. Additionally, the cooling curves of the alloy were obtained based on the measured data. 

Specimens for the metallographic examination were grinded through water matter paper, polished with 1.5 μm diamond grinding paste, and then etched with a solution of 1% hydrofluoric acid. Before observation of the three-dimensional morphology, a 10% sodium hydroxide solution was used to dissolve the Al matrix (0.5–1 h) in order to expose more of the Fe-phases embedded inside the Al matrix. The microstructures of Al-7Si-3Fe alloy were surveyed by a Zeiss Observer A1m optical microscope (OM), a Zeiss Evo MA10 scanning electron microscope (SEM, Carl Zeiss AG, Oberkochen, Germany) equipped with an Oxford X-Max 50 mm^2^ energy dispersive spectroscope (EDS, Oxford Instruments, Oxford, UK), and a Smartlab 9Kw X-ray diffraction (XRD, Rigaku Corporation, Tokyo, Japan). The samples were machined into a tensile testing rod with a diameter of 5 mm and a gauge length of 25 mm. The tensile tests were carried out at room temperature using an MTS model E45 testing machine at a loading velocity of 1 mm/s. The ultimate tensile strength (UTS) and elongation to fracture are the average values of at least three individual repeated tests. 

## 3. Results and Dicscussion

Figure 2a shows the XRD patterns of the Al-7Si-3Fe alloys. It is found that the as-cast and as-vibration Al-7Si-3Fe alloys consist of α-Al, Si, and iron-rich phases, especially the β-Al_5_FeSi phases in the as-cast alloy, which are replaced by the α-Al_8_Fe_2_Si phases in the as-vibration alloy. In addition, it is obvious that the iron phases in the as-cast alloy are elongated, while those in the as-vibration alloy are polygonal, as shown in Figure 2b,c. Subsequently, the element contents of phases A and B in Figure 2b,c are analyzed by EDS, and the results are shown in Table 2. It is found that the Fe/Si ratio is about 1 in the plate-like phase (A) and 1.9 in the polygonal bulk-like phase (B). According to the literature [24], the Fe/Si ratio is about 1 for β-Al_5_FeSi, and 1.9 for α-Al_8_Fe_2_Si in Al-Si-Fe alloys. Thus, it is confirmed that the plate-like phases in the as-cast alloy are β-Al_5_FeSi and the polygonal bulk-like phase in the as-vibration alloy is α-Al_8_Fe_2_Si (comparing Figure 2b,c). According to references [8,25], the difference in morphology of the iron-rich phase is due to the β phase belonging to the monoclinic lattice crystal structure, while the α phase is a hexagonal lattice.

Figure 3 shows OM graphs of as-cast and as-vibration Al-7Si-3Fe alloys and the detailed three-dimensional shapes of the β-Al_5_FeSi and α-Al_8_Fe_2_Si phases. As shown in Figure 3a, in the as-cast alloy, the microstructure consists of the developed α-Al dendrites and the thin and long plate-like β-Al_5_FeSi phases. The later ones are distributed throughout the matrix, which split it and led to a severe decrease in properties. In the as-vibration alloy, the coarse α-Al dendrites are refined, and the coarse dendrites transform into a fine and uniform rose-like grain, while the thin and long plate-like β-Al_5_FeSi phases are replaced by the polygonal, bulk-like α-Al_8_Fe_2_Si phases, as shown in Figure 3d. In addition, it is found that a small amount of short rod-like β-Al_5_FeSi phase is in the matrix (Figure 3d). The polygonal bulk-like phase and short rod-like phase are mainly distributed in the grain junction or grain boundaries (Figure 3d), which reduces the disadvantage of the properties of the alloy. To further understand the iron-rich phase, the detailed three-dimensional shapes are shown in Figure 3b,c,e,f. It is visible that the β-Al_5_FeSi phase exhibits a plate-like structure, with an average width of ~10 μm, especially, some plates cross each other and form a network structure, as Figure 3b,c. Dinnis et al. [26] has observed by serial sectioning that the needle-like β-Al_5_FeSi phase in the 2D radiographs is actually plate-like and the plate-like phases form complex and interconnected network structure each other. After vibration, the 3D morphology of iron-rich is mainly a polyhedral structure, as shown in Figure 3e,f. According to Gao [27], the 3D morphology of the blocky α-AlFeSi particles indicates an obvious polyhedral shape from the corresponding fractography image. The tensile tests of as-cast and as-vibration alloys show that after applying vibration, the ultimate tensile strength and elongation increased from 190.6 MPa and 1.6% to 220 MPa and 2.6%, respectively. 

Figure 4a indicates the solidification sequence of the Al-7Si-3Fe alloy based on Thermo-Calc software. The α-Al_8_Fe_2_Si phase first begins to crystallize at 658 °C, as shown in Figure 4a. As the temperature drops, the α-Al_8_Fe_2_Si phase continues to precipitate. When the temperature drops to 613 °C, the (Al) begins to form because of the decrease in Si and Fe in the liquid. As Si increases in the remaining melt and the temperature drops to 611 °C, the α-Al_8_Fe_2_Si phase begins to transform into the β-Al_5_FeSi phase and a quasi-peritectic reaction occurs (L + α-Al_8_Fe_2_Si → (Al) + β-Al_5_FeSi). When the temperature decreases to 574 °C, the liquid composition moves to the eutectic point, and the eutectic (Al), (Si), and β-Al_5_FeSi phases crystallize until all of the liquid is exhausted. These solidification reactions are consistent with those reported in References [8,9]. Figure 4b shows the cooling curves of the as-cast and as-vibration alloys measured by experiment. It can be seen from Figure 4b that the as-vibration melt has a higher cooling rate, and the cooling rate played a critical role in the control of the solidification structure. Liu [14] indicated the coarse iron-rich intermetallics were refined to a significant extent by increasing the cooling rate. In addition, Figure 4b suggests the α-Al_8_Fe_2_Si phase has earlier precipitation time under vibration, compared with traditional casting. 

As mentioned above, the plate-like β-Al_5_FeSi phases are substituted by the polygonal bulk-like α-Al_8_Fe_2_Si phases in an as-vibration alloy (as shown in Figure 2 and Figure 3). The following three reasons are responsible for the modification mechanisms and the schematic drawing in Figure 5. 

First, the vibration promotes the formation of free crystals. According to crystal dissociating theory [28], some grains formed on the mold wall and the cooling liquid surface are more easily free to melt under the forced convection, which promotes the formation of grain. Additionally, it is well known that forced convection is induced by vibrations in the melt. Therefore, based on the solidification sequence in Figure 4a, the first precipitation α-Al_8_Fe_2_Si phases move toward melt in forced convection, and then the new grains continue to form in the mold wall and the cooling liquid surface. As a result, a large number of α-Al_8_Fe_2_Si phases formed in the melt, as shown in Figure 5a.

Second, the vibration promotes heterogeneous nucleation. References [28,29] show that forced convection caused by vibration can eliminate the difference in temperature and composition field in the melt, and the whole melt is in an undercooling state. In this case, a large number of effective nucleation sites can carry out heterogeneous nucleation in the melt, which increases the heterogeneous nucleation rate of the α-Al_8_Fe_2_Si phase (as shown in Figure 5b). In addition, Que et al. [30] reported that the possible heterogeneous nucleation substrates for the primary iron-rich intermetallic compounds are the casting mould wall and the native and in-situ oxides. Similarly, some research [31,32] also suggested that iron-rich intermetallic compounds can nucleate on the outer surface of oxide bi-films and xoides, by observation of direct contact. As we all know, the oxides are easily formed on the melt surface during the solidification process of aluminum alloy, and then the oxides on the melt surface are continuously introduced into the melt by vibration, thus promoting heterogeneous nucleation. Meanwhile, the oxides formed on the mould wall are washed down by vibration and move toward melting, which promotes nucleation.

Third, the vibration can improve the growth rate. As we all know, the vibration can induce a high heat transfer inside the molten metal to the mold interface, which leads to a high cooling rate of the melt, as shown in Figure 4b. A large number of α-Al_8_Fe_2_Si phase nuclei formed previously grow rapidly at a higher cooling rate. Reference [33] shows that when the α-Al_8_Fe_2_Si achieves to a certain size, it is difficult to transform to the β-Al_5_FeSi through quasi-peritectic due to the hindrance of atomic diffusion (Figure 5c). However, not all α-Al_8_Fe_2_Si phase nuclei grow to a certain size rapidly under vibration; there are still some α-Al_8_Fe_2_Si phases that transfer to β-Al_5_FeSi phases by quasi-peritectic reaction, such as some short rod-like β-Al_5_FeSi phases in Figure 3d.

Above all, it is found that the quasi-peritectic reaction step of 614 °C disappears, as shown in Figure 4b, which indicates the quasi-peritectic reaction was hindered by vibration. Meanwhile, a lot of Fe and Si elements are consumed due to free crystals, and the high nucleation and growth rate of the α-Al_8_Fe_2_Si phase limit the eutectic reaction L → (Al) + β-Al_5_FeSi + Si. At the same time, Figure 4b shows that the eutectic reaction time of 574 °C decreases under vibration. According to the solidification sequence (Figure 4a), the β-Al_5_FeSi phase is formed mainly by quasi-peritectic reaction and eutectic reaction. As result, the plate-like β-Al_5_FeSi phases are difficult to crystallize during solidification. However, there is still a small number of β-Al_5_FeSi phases in the matrix, and these are refined to short rod-like, as shown in Figure 3d, due to the higher cooling rate under vibration. According to [34], the relationship between cooling rate and grain size is given by:(1)$$GS=a\times {CR}^{-b}$$

The value of *b* is in the range of 0.3–0.6. Thus, the calculated formula is:(2)$$GS=34.7\times {CR}^{-0.5}$$

Based on the above formula, the higher cooling rate can refine effectively grain. Meanwhile, the coarser α-Al dendrites are refined to a fine and uniform rose-like grain under vibration, as shown in Figure 3a,d. Therefore, applying vibration to the melt during solidification can effectively modify the iron-rich phase and refine the iron-rich phase and α-Al dendrites, which improve the properties of the alloy.

## 4. Conclusions

The modification behavior and mechanism of mechanical vibration on the iron-rich phase in an Al-7Si-3Fe alloy were investigated. The main findings are as follows:

(1) When the mechanical vibration (frequency: 30HZ, amplitude: 0.6 mm, direction: vertical) was applied on the Al-7Si-3Fe alloy during solidification, the coarse α-Al dendrites were refined to the fine and uniform rose-like grain, and the primary plate-like β-Al_5_FeSi transformed to polygonal bulk-like α-Al_8_Fe_2_Si. The ultimate tensile strength and elongation were increased to 220 MPa and 2.6%.

(2) The solidification sequence of Al-7Si-3Fe alloy based on the Thermal-Calc was L → α-Al_8_Fe_2_Si (658 °C), L → α-Al_8_Fe_2_Si+ (Al) (613 °C), quasi-peritectic reaction: L + α-Al_8_Fe_2_Si → β-Al_5_FeSi (611 °C), and eutectic reaction: L → (Al) + β-Al_5_FeSi+ Si (574 °C)

(3) When applying vibration during solidification, the cooling rate of the Al-7Si-3Fe alloy increases, and the quasi-peritectic reaction step of 614 °C disappears. This is mainly due to a forced convection and a higher heat transfer caused by vibration.

## Figures and Tables

**Figure 1 materials-16-01963-f001:**
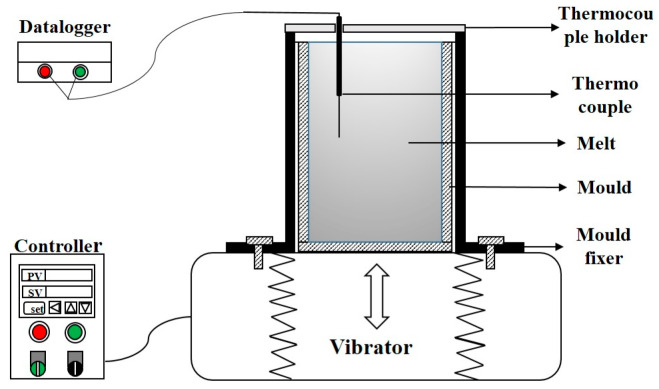
The schematic of a mechanical vibration device.

**Figure 2 materials-16-01963-f002:**
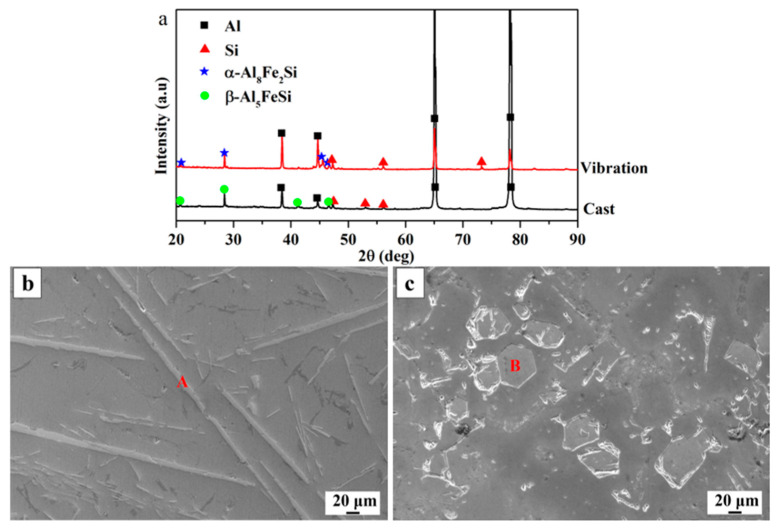
XRD (**a**) and SEM images of as-cast Al-7Si-3Fe alloy (**b**) and as-vibration (**c**).

**Figure 3 materials-16-01963-f003:**
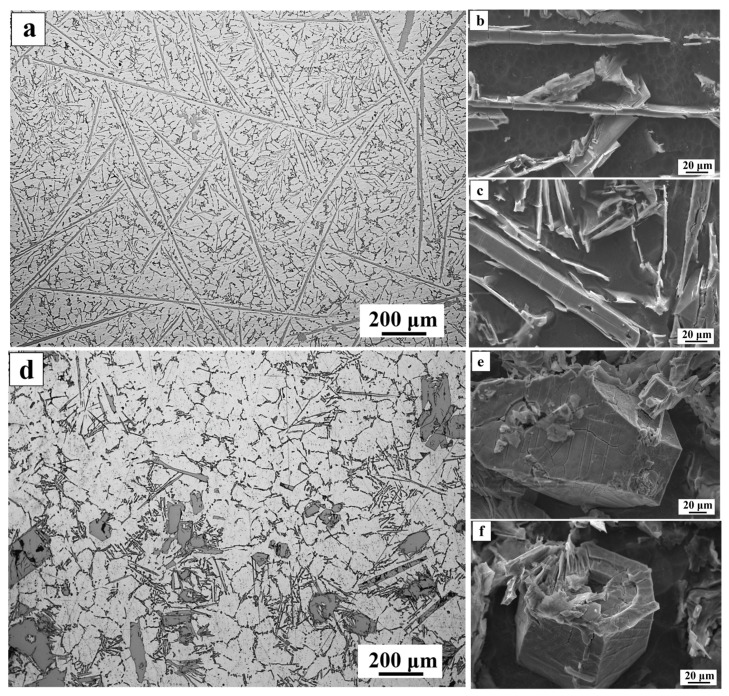
(**a**) OM of as-cast Al-7Si-3Fe alloy; (**b**,**c**) 3D morphologies of β-Al_5_FeSi phase; (**d**) OM of as-vibration Al-7Si-3Fe alloy; (**e**,**f**) 3D morphologies of α-Al_8_Fe_2_Si phase.

**Figure 4 materials-16-01963-f004:**
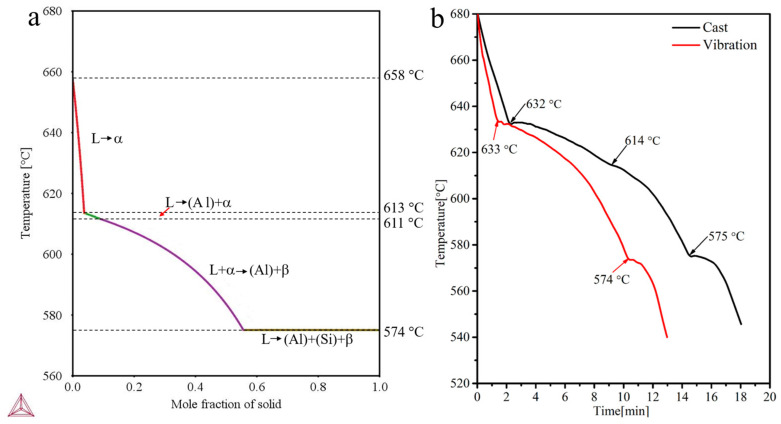
The solidification sequence of Al-7Si-3Fe based on Thermal-Calc software (**a**) and cooling curves of as-cast and as-vibration Al-7Si-3Fe alloys measured by experiment (**b**).

**Figure 5 materials-16-01963-f005:**
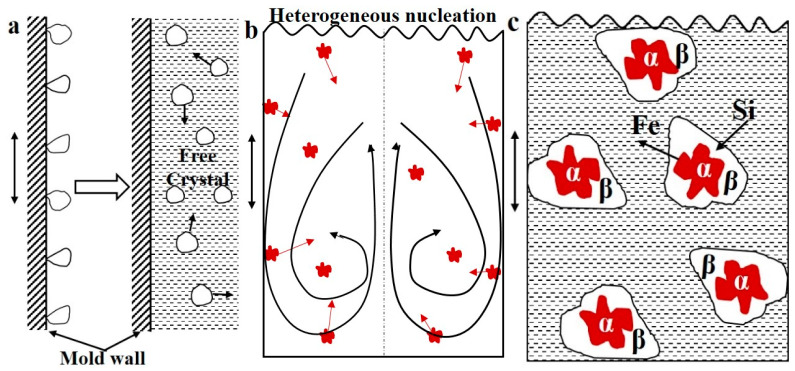
The schematic diagram of the transformation mechanism of iron-rich phase: (**a**) free crystal, (**b**) heterogeneous nucleation, and (**c**) element diffusion in quasi-peritectic reaction.

**Table 1 materials-16-01963-t001:** Chemical compositions of Al-Si-Fe alloys.

Alloy	Mass Fraction/%		
	Si	Fe	Al
As-cast	6.62	2.86	Bal.
As-vibration	6.47	2.87	Bal.

**Table 2 materials-16-01963-t002:** EDS results of the different iron-rich phases in Al-7Si-3Fe alloys.

Alloy	Phase	Atomic Percentage			Phase Formula
		Al	Si	Fe	
As-cast	A	71.02	15.69	13.29	β-Al_5_FeSi
As-vibration	B	70.42	10.14	19.44	α-Al_8_Fe_2_Si

## Data Availability

All data are available from the corresponding author on reasonable request.

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
