# Peer review of "Modification of Iron-Rich Phase in Al-7Si-3Fe Alloy by Mechanical Vibration during Solidification"

_materials, 2023, doi:10.3390/ma16051963_

Round 1

Reviewer 1 Report

Please review the notes in the pdf file and make the "major" edit.

Reviewer 2 Report

This paper is a study on the effect of mechanical vibration on the evolution of microstructure during solidification of aluminum alloy. The effect of mechanical vibration was explained by comparing the casting microstructure with the microstructure subjected to mechanical vibration. However, in my opinion, the depth of the content of the discussion of the author's claim is not sufficient to allow publication in this journal. Details are below.

1.     In the experimental method, there is no mention of the vibration frequency or amplitude mentioned in the conclusion. The experimental method should be described in more detail.

2.     Again, no information was given on whether the mechanical vibration was tested with only one frequency or several frequencies. It is judged that if the difference in solidification according to the vibration frequency or amplitude was compared, it would have shown better results. Overall, these areas are not discussed at all, so revision is required.

3.     On page 2, In this sentence, “The raw materials were pure Al, Al-20 %Si and Al-20 %Fe mater alloy.”, please indicate clearly whether % refers to weight fraction or atomic fraction.

4.     In 2. Experimental, please add detailed information about various test equipment (e.g. manufacturer, specifications, etc.)

5.     Figure 2 XRD image resolution is too low. It needs to be replaced with a high-resolution image.

6.     The scale bar in Fig. 3(c) is hidden. The figure needs correction. Also, please indicate clearly whether the scale bars in Fig. 3(a) and Fig. 3(d) are the same size.

7.     In Fig.4, Please state exactly whether the cooling curve calculation in thermo-calc is Scheil cooling or equilibrium cooling. This is because the analysis may differ depending on which cooling calculation is used.

Typo

Line 7 on Page 2, 100HZà 100Hz (change uppercase to lowercase)

On page 2, Al-20 % Fe mater alloyà Al-20 % Fe master alloy (materàmaster)

On page 2, 10% Sodium hydroxide à 10% sodium hydroxide (change uppercase to lowercase)

On page 4, elongation increased form 190.6 MPa,à elongation increased from 190.6 MPa, (formàfrom)

Reviewer 3 Report

Dear authors,

The main result is formulated in conclusion number 1:

 "When mechanical vibration (frequency: 30 Hz, amplitude: 0.6 mm) was applied...". In my opinion, a remarkable experimental result has been obtained based on the use of low-frequency elastic waves. However, when there is an interpretation, the explanation lies in the occurrence of convection in the melt. Therefore, questions arose.

1. Under what conditions calculations were made Fig.4, b.

2. Can you estimate the degree of convection and its effect on thermal conductivity.

3. Another mechanism is considered as an alternative to convection. For example, the difference between low-frequency and ultrasonic treatments methods of alloys are discussed in paper I. E. Ignat’ev at al., 2015.

I. E. Ignat’ev, E. A. Pastukhov, E. V. Ignat’eva. Principal distinction of the methods of low-frequency and ultrasonic effects on Melts. Russian Journal of Non-Ferrous Metals. 2015. 55(6):509-512. DOI: 10.3103/S1067821214060091

Round 2

Reviewer 1 Report

Thank you for the changes and edits made. It is acceptable in its final form.

Reviewer 2 Report

This paper is suitable for the publication in this journal.